# Stem Cell Theory of Cancer: Rude Awakening or Bad Dream from Cancer Dormancy?

**DOI:** 10.3390/cancers14030655

**Published:** 2022-01-27

**Authors:** Shi-Ming Tu, Marcos R. Estecio, Sue-Hwa Lin, Niki M. Zacharias

**Affiliations:** 1Department of Genitourinary Medical Oncology, The University of Texas MD Anderson Cancer Center, Houston, TX 77030, USA; 2Department of Epigenetics and Molecular Carcinogenesis, The University of Texas MD Anderson Cancer Center, Houston, TX 77030, USA; mestecio@mdanderson.org; 3Department of Translational Molecular Pathology, The University of Texas MD Anderson Cancer Center, Houston, TX 77030, USA; slin@mdanderson.org; 4Department of Urology, The University of Texas MD Anderson Cancer Center, Houston, TX 77030, USA; NMZacharias@mdanderson.org

**Keywords:** cancer dormancy, cancer stem cells, unified theory, clonal origin, seed and soil, prolonged remission, late relapse, second malignancy, stress

## Abstract

**Simple Summary:**

The stem cell theory of cancer predicates that both normal and cancer stem cells can be induced into or released from dormancy depending on their multipotential constraints and microenvironment restraints. A unified theory of cancer predicts that even though genetic makeup in cancer dormancy maybe pivotal, cellular context must be paramount. After all, both normal stem cells and cancer stem cells proliferate and differentiate. They can be stationary and migratory. They can be static and dynamic. Importantly, gonadal germ cells are prototype stem cells and germ cell tumor of the testis (TGCT) is a model stem cell cancer. In a TGCT and other cancers, we witness the many manifestations and myriad revelations of cancer dormancy in prolonged remissions, late relapses, second malignancies, and fulminant cancers.

**Abstract:**

To be dormant or not depends on the origin and nature of both the cell and its niche. Similar to other cancer hallmarks, dormancy is ingrained with stemness, and stemness is embedded within dormancy. After all, cancer dormancy is dependent on multiple factors such as cell cycle arrest, metabolic inactivity, and the microenvironment. It is the net results and sum effects of a myriad of cellular interactions, interconnections, and interplays. When we unite all cancer networks and integrate all cancer hallmarks, we practice and preach a unified theory of cancer. From this perspective, we review cancer dormancy in the context of a stem cell theory of cancer. We revisit the seed and soil hypothesis of cancer. We reexamine its implications in both primary tumors and metastatic lesions. We reassess its roles in cell cycle arrest, metabolic inactivity, and stemness property. Cancer dormancy is particularly revealing when it informs us about the mysteries of late relapse, prolonged remission, and second malignancy. It is paradoxically rewarding when it delivers us the promises and power of cancer prevention and maintenance therapy in patient care.

## 1. Introduction

When we formulate a stem cell theory of cancer, dormancy may be one of the more enlightening biological mechanisms to demonstrate the appeal and application of this theory. After all, dormancy is deeply rooted in the seed and soil hypothesis [1] and ingrained in the process of germination and dissemination of cancer. A cancer stem cell is similar to a seed that remains dormant when the niche is not right.

When we consider a unified theory of cancer, dormancy may be one of the more illuminating examples to illustrate the prevalence and pertinence of this theory. After all, cancer dormancy is dependent on multiple factors such as cell cycle arrest, metabolic inactivity, and the microenvironment [2,3,4,5].

From this perspective, we review cancer dormancy in the context of a stem cell theory of cancer (Figure 1). We revisit the seed and soil hypothesis of cancer. We reexamine its implications in both primary tumors and metastatic lesions. We reassess its role in cell cycle arrest, metabolic inactivity, and stemness property. Cancer dormancy is particularly revealing when it informs us about the mysteries of late relapse, prolonged remission, and second malignancy. It is paradoxically rewarding when it delivers us the promises and power of cancer prevention and maintenance therapy in patient care. When we empower ourselves with the right cancer theory and equip ourselves with the proper scientific method, we will spare ourselves more bad dreams or rude awakenings in cancer care.

## 2. Seed and Soil

In 1889, Stephen Paget published that breast cancer metastasized to certain organ sites (e.g., liver) far more often than others (e.g., spleen) [1]. To account for this salient observation, he proposed the classic seed and soil hypothesis. Importantly, this seminal hypothesis may explain another key cancer hallmark besides metastasis, namely dormancy. In fact, dormancy is innately ingrained in a seed, and is intimately influenced by the soil. A seed can remain dormant forever. It is similar to a normal or cancer stem cell that can renew and resume in perpetuity. The very being of a seed is dependent on the soil. This is the very essence of a stem cell theory of cancer.

In many respects, the idea if not the observation of dormancy contradicts a genetic theory of cancer. Although genetic mutations may be obligatory in the initiation and promotion of cancer, they may not contribute to the pathogenesis of cancer dormancy. After all, genetic mutations are less likely to occur and unlikely to be selected for in the absence of any cellular growth, division, or activity [6]. Since dormancy is an intrinsic property of stem cells (and certain somatic cells), it may not be necessary to invoke any specific mutations for those cells to be induced into or released from dormancy. The discovery that cells with identical genetic background can display variable functional phenotypes, including dormancy [7], and cells with certain integrin profiles have unique dormancy potential without regard to any specific genetic defects [8] is also in conflict with the genetic theory of cancer.

To be dormant or not depends on the nature and status of both the cell and its microenvironment [9]. Paracrine factors at the site of metastasis, such as bone morphogenetic proteins, may induce dormancy in metastasis-initiating cells by inhibiting their capacity to self-renew [10]. Conversely, metastasis-initiating cells (similar to their adult stem cell counterparts) secrete Coco (an antagonist of transforming growth factor (TGF)-beta ligands), which reactivates dormant breast cancer cells in the lung [11]. Similarly, periostin, another matrix protein secreted by stromal fibroblasts in response to TGF-beta in stem cell niches, promotes activation from dormancy by facilitating the presentation of Wnt ligands to tumor cells [12]. Thus, dormancy reaffirms a fundamental principle of cancer: the malignant phenotype can be both dynamic and static and is intricately dependent on a close interplay between the involved cell and its niche.

## 3. Dormancy

A proper definition enables us to effectively communicate about cancer dormancy. Cellular dormancy is nonproliferation, when cells are under cell cycle arrest. There are three types of nonproliferating cells: dormant, quiescent, and senescent. Both quiescent and dormant cells have the capacity to re-enter the cell cycle. Quiescent cells are in a temporary pause in proliferation, while dormant cells are in a persistent arrest of nonproliferation. Senescent cells have lost the capacity to re-enter the cell cycle. They are metabolically active but proliferatively incapacitated.

We propose a stem cell theory to elucidate the origin and nature of cancer dormancy. A stem cell theory may be the unifying theory that connects all the pieces of the puzzle of cancer, including cancer dormancy (Figure 1): low levels of proliferation-associated proteins, lack of apoptotic markers, and involvement of dormancy-regulating and -inducing signaling cascades (such as TGF-beta pathways, reduced PI3K/AKT signaling). It embraces many other aspects of cancer dormancy, including mitigated MHC-I expression [13] and attenuated metabolic signaling through hypoxia/TGF-beta [14], as well as through mTOR [15] or MYC oncogenic pathways [16]. Importantly, a unifying theory of cancer dormancy would encompass immune/inflammatory factors [17] and autophagy components [18], and pluripotency through NR2F1 and NANOG [19], epithelial-to-mesenchymal transition (EMT) through LOXL2 [20] and PRRX1 [21], all of which are in some ways directly or indirectly related or linked to stemness.

## 4. Stemness

When stemness is an overt characteristic of cancer, it could be the code to decipher a conundrum of cancer dormancy. In many respects, stemness accounts for all the conventional hallmarks of cancer—autonomy, metastasis, heterogeneity, immune evasion, and genetic instability [22,23]. When stemness is a preeminent feature of cancer, it alludes to a stem cell origin and stem-like nature of cancer.

Similar to other cancer hallmarks, dormancy is ingrained in stemness, and stemness is embedded within dormancy. A prime example of a dormant stem cell is an inactive germ cell. Without fertilization, an ovum may remain dormant for the rest of the host’s life span. After fertilization, it will form all the diverse organs and tissues in an offspring. It is of interest that transformed fetal gonocytes in the form of intratubular germ cell neoplasia unclassified (IGCNU) [24] remain dormant and do not become malignant until after puberty a decade or several decades later.

There is irony in stemness and dormancy. Stemness is good because regeneration and wound healing is good. However, stemness is also bad when it is prevalent in malignancy. Dormancy is good when a cancer is in slumber, but it is also bad because the cancer may wake up. Hence, a putative dormancy factor, TGF-beta, is both a tumor suppressor, because it mediates anti-proliferative and apoptotic effects, and a tumor promoter, because it induces tumor motility, metastasis, and EMT. TGF-beta is a different actor in a fetus vs. an adult, during embryogenesis vs. carcinogenesis. What it does in the fetus may be perfectly benign during embryogenesis, but the same activity in an adult may be patently malignant during carcinogenesis.

In many respects, stemness is an epiphany of oncology recapitulating ontogeny. The exact role and proper function of a normal stemness or a malignant stem-like factor is dependent on the exact or proper context, i.e., timing and placement. If cancer has a stem cell origin and is a stem cell disease, it does not need to reprogram or reinvent its stemness. It does not need to hijack or retrieve what it already owns.

## 5. Primary vs. Metastatic Cancer

A stem cell origin of cancer can better account for dormancy and metastasis in the seed and soil for primary and metastatic tumors than a genetic origin of cancer does. After all, once a primary tumor forms, there are not many more driver mutations that appear and accumulate [25]. A multipotent progenitor cell is more capable of spreading to and colonizing various sites than an oligopotent or unipotent progenitor cell. They provide and receive different sets of cues from their respective microenvironments to become exuberant or remain dormant.

The observation that primary tumors and metastatic lesions are different diseases and can be modified by specific microenvironmental factors has far-reaching biological implications and clinical applications. For example, the 21-gene recurrence score (Oncotype Dx) predicts risk of local breast cancer recurrence (ipsilateral breast, chest wall, and regional nodal) regardless of systemic disease control with tamoxifen or chemotherapy [26]. After induction therapy, the maintenance and prolongation of remission by targeting minimal residual disease (i.e., cancer stem cells) and the bone microenvironment (e.g., with anti-inflammatory and anti-metabolic agents) may improve clinical outcome regardless of local disease control in the past or in the future [27].

Importantly, when a primary tumor or its metastatic lesions are inherently stable or dormant, surgical extirpation with curative intent of the former may be feasible and with improved control of the latter may be appropriate. Turajlic et al. [28] demonstrated that primary renal cell carcinoma (RCC) with low intratumoral heterogeneity (ITH), but elevated somatic copy-number alterations (SCNAs) had rapid progression at multiple sites. However, those with low ITH and low fraction of the genome affected by SCNAs had overall low metastatic potential. Results from their study suggest that the removal of primary tumor is beneficial for those indolent RCC with high ITH (even in the presence of metastasis) but not for those fulminant RCC with low ITH and high SCNAs (despite absence of metastasis).

Interestingly, low ITH high SCNAs reflect aneuploidy and implicate aberrant asymmetric division, suggesting that a stemness origin may be involved in the evolution of this malignant phenotype [29]. In principle, when we have treatments that keep a primary tumor or its metastatic lesions stable or dormant, surgery may be beneficial even in the setting of metastatic disease. In practice, a stem cell theory of dormancy may improve our design of adjuvant therapy and our selection of patients for metastasectomy.

## 6. Prolonged Remissions

Sometimes, disease outliers provide invaluable clues about cancer dormancy. Exceptional cases of extreme clinical outcomes (both good and bad) may unveil fundamental mechanisms of action in cancer dormancy.

When an active, threatening cancer calms down and becomes dormant, something must be keeping the molten lava bottled up, as in a dormant volcano, or the husky body bundled up, as in a hibernating bear. What physical barriers delay a volcano from erupting? What sleeping potions enable the bear to keep slumbering?

Prolonged remission is particularly intriguing in a supposedly deadly metastatic cancer when we do not expect durable remission or anticipate extended survival. We are not talking about patients with indolent tumors, such as follicular lymphomas or chronic lymphocytic leukemias, who may live for years if not decades without or despite treatment.

For example, Greenberg et al. showed that about 20% of patients with metastatic breast cancer who had experienced a complete remission after treatment would live beyond ten years, perhaps even twenty years [30]. Surprisingly, even some patients with an incomplete remission could experience a prolonged remission. It seems that a small proportion of patients with other metastatic cancers, such as renal cell carcinoma and melanoma, may also attain prolonged remissions [31,32].

A stem cell theory of cancer is consistent with and predicts the occurrence of prolonged remissions. It is conceivable that in any minimal residual disease after induction systemic treatments, the drug-resistant, non-cycling cells comprise cancer stem cells. It is plausible that when we do not trigger or rattle those putative cancer stem cells, they will remain quiescent, if not dormant, for a prolonged period, if not permanently.

## 7. Very Late Recurrence

The unique occurrence of very late recurrence is particularly noteworthy in the context of a stem cell theory of cancer [33]. In most instances, the primary tumor has been extirpated. Therefore, the recurrent disease is likely related to micrometastases that eventually become reactivated and insurrected.

Approximately 1% of patients with germ cell tumor of the testis (TGCT) develop very late recurrence more than 5 years after diagnosis [34]. The question is: What is unique about this 1% of TGCT that causes very late recurrence, and what can be done to prevent it?

Moore et al. [35] examined a group of 25 TGCT patients with very late recurrence. The median time for relapse was 16.1 years. The longest time was 33.1 years. All recurrent tumors comprised somatically transformed tumor, yolk sac tumor, and/or teratoma.

The irony of cancer dormancy and very late recurrence is that what were considered favorable tumor phenotypes before the era of chemotherapy—indolent TGCT such as yolk sac tumor and teratoma—have become unfavorable entities since the inception of chemotherapy (in the 1970s). In 1946, about 90% of patients with metastatic TGCT died within one year of diagnosis [36]. Today, over 90% of patients with the same diagnosis are cured [37]. Unfortunately, many of the patients who still die from TGCT harbor indolent chemotherapy-resistant TGCT, such as yolk sac tumor and teratoma [38,39]. 

In many respects, the indolent and dormant nature of yolk sac tumor and teratoma belies their treacherous eventual outcome. Importantly, ensconced within a residual, supposedly innocuous teratoma after chemotherapy are dormant and covert progenitor stem cells that may reawaken and relapse as a yolk sac tumor and/or a somatically transformed tumor [40]. 

There is a reason certain teratomas are malignant and need to be removed to be cured. It is not the teratoma per se that is malignant. It is the dormant, aberrant progenitor stem cells lodged within the teratoma that make it malignant and necessary to be removed in a timely manner.

A stem cell theory of TGCT may empower us to prevent very late recurrence and enable us to cure the remaining 10% of TGCT patients who still die from this otherwise very curable cancer.

## 8. Second Malignancies

The fact that cancer may lay dormant and resurface years if not decades later alludes to a stem cell theory of cancer. When the re-emergent malignancy is related to the first in its genotype but not phenotype, it reaffirms if not confirms a stem cell theory of cancer.

Umbreit et al. [40] found that about half of a variety of subsequent malignant neoplasms (SMN), including lung cancer, colon cancer, and kidney cancer, in 42 patients with a previous history of TGCT contained a common genetic biomarker for TGCT, namely isochromosome 12p [i(12p)] (12%) or 12p gain (38%), by fluorescent in situ hybridization analysis. These SMN comprising non-TGCT occurred about 18 years (range 0.4–53.6 years) after an initial diagnosis of TGCT.

When a malignant tumor is capable of dormancy for years if not decades, its cell of origin must be perennial rather than ephemeral [41]. If it has the capacity to undergo somatic transformation into different cancer types and subtypes, the cell of origin must be multipotent and capable of differentiation into various cellular lineages. Had Umbreit et al. demonstrated that different and separate SMN in both space and time from the same patient did harbor a similar if not identical genetic profile (by whole-exome sequencing), the biological and clinical implications would be more than profound. Importantly, had they elaborated that an SMN and a subsequent benign neoplasm from the same patient also shared a similar if not the same genetic profile, a stem cell origin of cancer would be beyond dispute.

## 9. Stress and Dormancy

Although we suspect that stress is a culpable process that activates or releases cancer from dormancy, it is a difficult entity to incriminate, in part because it seems utterly intangible and unmeasurable.

Nevertheless, there is growing evidence that the nervous system could be complicit in cancer’s growth and spread, i.e., cancer-nerve (e.g., β-adrenergic) crosstalk. There are gripping data suggesting that chronic/excessive stress abets cancer progression and that targeting or interrupting this link may slow cancer’s growth and stop its spread.

For example, patients on a β-blocker experience better cancer-specific survival [42] and longer median overall survival (90.0 vs. 38.2 months; *p* < 0.001) [43]. It would be remarkable indeed if β-adrenergic signaling pathways could somehow affect stemness networks and could be harnessed for therapeutic purposes in cancer care.

Gross et al. [44] discovered that an antidepressant could also be an anticancer agent. They found that an antidepressant, phenylzine (a monoamine oxidase inhibitor, MAOi), elicited a PSA response in 55% of patients with prostate cancer. MAO-A is a mitochondria-bound enzyme that induces EMT through activation of VEGF and its coreceptor neurophilin-1 and stabilizes HIF1-alpha. MAO-A-dependent activation of neurophilin-1 promoted AKT/FOXO1/TWIST1 signaling. MAO-A also catalyzes the degradation of monoamine neurotransmitters (norepinephrine, epinephrine, serotonin, and dopamine) and dietary amines by oxidative deamination, which produces hydrogen peroxide, a major source of reactive oxygen species (ROS), as a by-product. ROS cause tumor initiation and progression through DNA damage in cancer cells [45].

Perego et al. [46] demonstrated that stress hormones cause a rapid release of proinflammatory S100A8/A9 proteins by neutrophils in dormancy models of lung and ovarian cancers. S100A8/A9 induces the activation of myeloperoxidase, resulting in the accumulation of oxidized lipids in those cells. Upon release from neutrophils, the oxidized lipids upregulate the fibroblast growth factor pathway in tumor cells and cause the tumor cells to exit from dormancy and form new tumor lesions. Higher serum concentrations of S100A8/A9 were associated with shorter time to recurrence in patients with lung cancer after complete tumor resection.

Therefore, we should take heed that stress may awaken cancer cells from dormancy. Perhaps we have been ignorant that certain natural products that possess cancer prevention properties may also provide cancer dormancy benefits. For example, curcumin (derived from turmeric) is one of only about 40 promising agents out of over 1000 tested by the NCI (since 1987) with known cancer prevention capabilities—promoting apoptosis, inhibiting survival signals, scavenging ROS, and reducing the inflammatory cancer microenvironment [47]. What is not known is that curcumin is also an antidepressant (i.e., it is a MAOi) and an anxiolytic (i.e., it may reduce stress) [48,49]. It is also unknown whether curcumin may owe some of its cancer prevention and cancer dormancy benefits to its versatile functions and multifaceted actions, including anti-inflammation, anti-depression, and anti-stress.

## 10. Conclusions

The stem cell theory of cancer predicates that both normal and cancer stem cells can be induced into or released from dormancy depending on their multipotential and microenvironmental constraints (Figure 1). A unified theory of cancer predicts that even though genetic makeup in cancer dormancy may be pivotal, cellular context is paramount. We anticipate that the stem cell theory of cancer is compatible with the seed and soil hypothesis. We expect that primary tumors are distinct from their metastatic counterparts because of their separate onco-niches and disparate phenotypes, if not genotypes. After all, both normal stem cells and cancer stem cells proliferate and differentiate. They can be stationary and migratory. They can be static and dynamic. Importantly, gonadal germ cells are prototype stem cells. TGCT is a model stem cell cancer—it can be floridly active and latently dormant, which is similar to its benign germ cell counterpart. In a TGCT and other cancers, we witness the many manifestations and myriad revelations of cancer dormancy in prolonged remissions, late relapses, and second malignancies. Perhaps when we look inside and peer into the intricacies of cancer dormancy, we will awaken to cancer’s stem cell origin and nature. When we look outside and view beyond the horizons of our conventions and traditions, perhaps we will begin to dream more about cancer cures and prolonged remissions and dread less about cancer nightmares related to late relapses, second malignancies, or fulminant cancers.

## Figures and Tables

**Figure 1 cancers-14-00655-f001:**
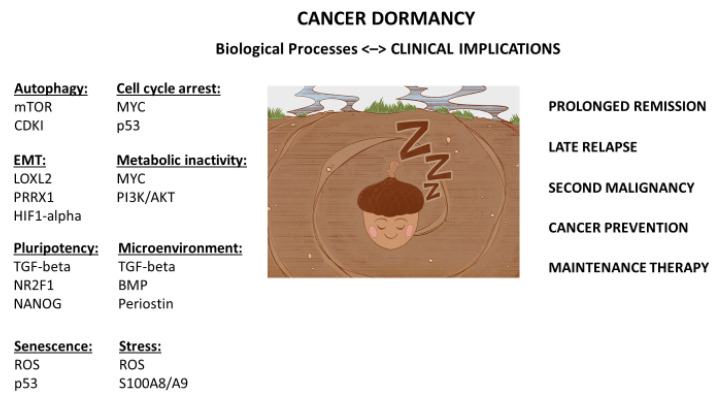
Unified theory and stem cell origin of cancer dormancy: biological processes and clinical implications. Illustration by Benjamin Tu.

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
