# Peer review of "Stem Cell Theory of Cancer: Rude Awakening or Bad Dream from Cancer Dormancy?"

_cancers, 2022, doi:10.3390/cancers14030655_

Round 1

Reviewer 1 Report

The authors reviewed cancer dormancy in the context of a stem cell theory of cancer by revisiting the seed and soil hypothesis of cancer. They discussed comprehensively and thoroughly the aspects that are relevant to cancer dormancy. The reviewer believe that this review will be useful not only for the clinicians of cancer treatment, but also for cancer research specialists. There are very few minor issues to be addressed.

Minor

  1. Figure 1 summarizes important keywords for cancer dormancy in terms of biological processes and clinical significance. If possible, drawing the figure relating to each function of biological factors would be helpful for readers to understand.

  1. Lines 223 to 224

… necessary to remove in a timely manner.

… necessary to be removed in a timely manner.

Author Response

We thank the reviewers for your insightful comments and suggestions. We have provided our responses below in bold with changes highlighted in red underneath the reviewer comments. We have incorporated all changes recommended by the reviewers with the changes highlighted in red in the manuscript.

The authors reviewed cancer dormancy in the context of a stem cell theory of cancer by revisiting the seed and soil hypothesis of cancer. They discussed comprehensively and thoroughly the aspects that are relevant to cancer dormancy. The reviewer believe that this review will be useful not only for the clinicians of cancer treatment, but also for cancer research specialists. There are very few minor issues to be addressed.

We thank the reviewer for your insightful comments!

Minor

  1. Figure 1 summarizes important keywords for cancer dormancy in terms of biological processes and clinical significance. If possible, drawing the figure relating to each function of biological factors would be helpful for readers to understand.

Modified: Relating biological factors to function and separating biological processes on the left from clinical implications on the right side of the figure.

  1. Lines 223 to 224

… necessary to remove in a timely manner.

… necessary to be removed in a timely manner.

Done.

Reviewer 2 Report

In this review, the authors put forth the concept that cancer cell dormancy is related to the stem cell theory of cancer, and that this unified theory explains phenomenon such as metastasis, late recurrence, and secondary malignancies.

There are some issues that need to be addressed.

  1. The manuscript is filled with hyperbole and anthropomorphism which distract from the data. Please revise.
  2. The introduction is too lengthy and the “unified theory of cancer “ that they are trying to describe is not clearly introduced until the third paragraph. Need to shorten the introduction and make it more succinct.
  3. In second section, again very long and not to the point. What did Coco [11] show? It would be much more helpful to the reader to actually tell us rather than just “e.g.”.
  4. Third section is also too long and filled with fluff. The only important part of this section is the third paragraph. Should state what the pertinent hypothesis is, not just that there is one (line 96).
  5. Same issue with section four. Line 126 “pertinent hypothesis” but don’t state it and don’t provide much data backing their hypothesis in the section.
  6. Section 5, first and third paragraphs are just fluff and should be eliminated or highly edited. Say same thing stated in the first three pages of the manuscript. Lines 167, need to tell the readers that Oncotype Dx that they are explaining applies to breast cancer.
  7. In section 8, third and fourth paragraphs say the same thing and should be combined and shortened.
  8. Section 9, the entire section provides absolutely no evidence of stem cell theory in fulminant cancer. Just anecdotes and hyperbole.
  9. The manuscript should be revised to concisely provide data to prove their hypothesis of cancer cell dormancy is related to the stem cell theory of cancer.

Author Response

We thank the reviewers for your insightful comments and suggestions. We have provided our responses below in bold with changes highlighted in red underneath the reviewer comments. We have incorporated all changes recommended by the reviewers with the changes highlighted in red in the manuscript.

In this review, the authors put forth the concept that cancer cell dormancy is related to the stem cell theory of cancer, and that this unified theory explains phenomenon such as metastasis, late recurrence, and secondary malignancies.

There are some issues that need to be addressed.

  1. The manuscript is filled with hyperbole and anthropomorphism which distract from the data. Please revise.

Thank you for your comments and advice! We have either removed extraneous phrases or revised sections where data is lacking that weaken our overall arguments and distract from the main thesis of this article.

  1. The introduction is too lengthy and the “unified theory of cancer “ that they are trying to describe is not clearly introduced until the third paragraph. Need to shorten the introduction and make it more succinct.

Deleted, 1st paragraph: “When it grows into a multicellular plant with trunk, branches, and leaves, its multipotentiality (self-renewal) and multispecialty (differentiation) ensure perpetuity and durability.”

Deleted, 2nd paragraph: It is the net result and sum effect of myriad cellular interactions, interconnections, and interplays. When we unite all cancer networks and integrate all cancer hallmarks, we practice and preach a unified theory of cancer.”

  1. In second section, again very long and not to the point. What did Coco [11] show? It would be much more helpful to the reader to actually tell us rather than just “e.g.”.

Clarified: “Conversely, metastasis-initiating cells (like their adult stem cell counterparts) secrete Coco (an antagonist of TGF-beta ligands) that reactivates dormant breast cancer cells in the lung [11].

  1. Third section is also too long and filled with fluff. The only important part of this section is the third paragraph. Should state what the pertinent hypothesis is, not just that there is one (line 96).

Amended: “We propose a stem cell theory to elucidate the origin and nature of cancer dormancy. A stem cell theory may be the unifying theory that connects all the pieces of the puzzle of cancer, including cancer dormancy: low levels of proliferation-associated proteins, lack of apoptotic markers, and involvement of dormancy-regulating and -inducing signaling cascades (such as TGF-beta pathways, reduced PI3K/AKT signaling).”

  1. Same issue with section four. Line 126 “pertinent hypothesis” but don’t state it and don’t provide much data backing their hypothesis in the section.

Deleted, 3rd paragraph: to keep the article succinct as recommended, since “pertinent hypothesis” has been addressed in the previous section.

  1. Section 5, first and third paragraphs are just fluff and should be eliminated or highly edited. Say same thing stated in the first three pages of the manuscript. Lines 167, need to tell the readers that Oncotype Dx that they are explaining applies to breast cancer.

Deleted, 1st and 3rd paragraphs: to keep the article succinct as recommended. Agree that there is redundancy with respect to the seed and soil hypothesis.

Amended: “predicts risk of local breast cancer recurrence (ipsilateral breast, chest wall, and regional nodal) …”

Added: another example of the disparate effects of dormancy on primary vs metastatic kidney cancer.

“Importantly, when a primary tumor or its metastatic lesions are inherently stable or dormant, surgical extirpation with curative intent of the former may be feasible and with improved control of the latter may be appropriate. Turajlic et al. [28] demonstrated that primary renal cell carcinoma (RCC) with low intratumoral heterogeneity (ITH) but elevated somatic copy-number alterations (SCNAs) had rapid progression at multiple sites. However, those with low ITH and low fraction of the genome affected by SCNAs had overall low metastatic potential. Results from their study suggest that removal of primary tumor is beneficial for those indolent RCC with high ITH (even in the presence of metastasis), but not for those fulminant RCC with low ITH and high SCNAs (despite absence of metastasis).

Interestingly, low ITH high SCNAs reflect aneuploidy and implicate aberrant asymmetric division, suggesting that a stemness origin may be involved in the evolution of this malignant phenotype [29]. In principle, when we have treatments that keep a primary tumor or its metastatic lesions stable or dormant, surgery may be beneficial even in the setting of metastatic disease. In practice, a stem cell theory of dormancy may improve our design of adjuvant therapy and our selection of patients for metastasectomy. 

  1. In section 8, third and fourth paragraphs say the same thing and should be combined and shortened.

Combined and shortened: “When a malignant tumor is capable of dormancy for years if not decades, its cell of origin must be perennial rather than ephemeral [39]. If it has the capacity to undergo somatic transformation into different cancer types and subtypes, the cell of origin must be multipotent and capable of differentiation into various cellular lineages. Had Umbreit et al. demonstrated that different and separate SMN in both space and time from the same patient did harbor a similar if not identical genetic profile (by whole-exome sequencing), the biological and clinical implications would be more than profound. Importantly, had they elaborated that an SMN and a subsequent benign neoplasm from the same patient also shared a similar if not the same genetic profile, a stem cell origin of cancer would be beyond dispute.

  1. Section 9, the entire section provides absolutely no evidence of stem cell theory in fulminant cancer. Just anecdotes and hyperbole.

Deleted, section 9: to keep the article succinct as recommended. Agree that there may be other mechanisms that could account fulminant cancers beside dormancy factors.

  1. The manuscript should be revised to concisely provide data to prove their hypothesis of cancer cell dormancy is related to the stem cell theory of cancer.

We thank the reviewer for your astute recommendations, which have enhanced the content and context of this manuscript.

We have revised the manuscript to concisely provide data and by focusing on “Prolonged Remissions”, “Very late Recurrence”, “Second Malignancies”, and “Stress and Dormancy”, to support the hypothesis that cancer cell dormancy is related to the stem cell theory of cancer. We have also modified the section on “Primary vs Metastatic Cancer” and deleted the section on “Fulminant Cancers”, to streamline the overall argument and strengthen the main thesis of this article.

Round 2

Reviewer 2 Report

the queries have been addressed